# Monitoring the Evolution of the Aroma Profile of Lager Beer in Aluminium Cans and Glass Bottles during the Natural Ageing Process by Means of HS-SPME/GC-MS and Multivariate Analysis

**DOI:** 10.3390/molecules28062807

**Published:** 2023-03-20

**Authors:** Ana Carolina de Lima, Laura Aceña, Montserrat Mestres, Ricard Boqué

**Affiliations:** 1Chemometrics, Qualimetrics and Nanosensors Group, Department of Analytical Chemistry and Organic Chemistry, Universitat Rovira i Virgili, Campus Sescelades, 43007 Tarragona, Catalonia, Spain; 2Instrumental Sensometry Group (iSens), Department of Analytical Chemistry and Organic Chemistry, Universitat Rovira i Virgili, Campus Sescelades, 43007 Tarragona, Catalonia, Spain

**Keywords:** aluminium can, glass bottle, natural ageing, sensory analysis, packaging, HS-SPME/GC-MS, multivariate analysis

## Abstract

Headspace solid-phase microextraction coupled to gas chromatography-mass spectrometry (HS-SPME/GC-MS), sensory evaluation, and multivariate analysis were applied to monitor and compare the evolution of the aromatic profile of a lager beer in different types of containers (aluminum cans and glass bottles) during the natural ageing process. Samples were aged naturally for a year in the absence of light with a controlled temperature of around 14 °C +/− 0.5 °C. The sensory evaluation applied was a blind olfactometric triangle test between canned and bottled samples at different periods of aging: fresh, 6 months, and 11 months. The sensory evaluation showed that the panelists were able to differentiate between samples, except for the fresh samples from the brewery. A total of 34 volatile compounds were identified using the HS-SPME/GC-MS technique for both packaging types in this experiment. The application of multivariate analysis to the GC-MS data showed that the samples could not be differentiated according to the type of packaging but could be differentiated by the ageing time. The results showed that the combination of sensory, HS-SPME-GC-MS, and multivariate analysis seemed to be a valuable tool for monitoring and identifying possible changes in the aroma profile of a beer during its shelf life. Furthermore, the results showed that storing beer under optimal conditions helped preserve its quality during its shelf life, regardless of the type of packaging (aluminum can and glass bottle).

## 1. Introduction

The freshness of a beer is a key factor in determining its quality and is directly associated with the contents of aromatic compounds in the final product. From the moment beer is packaged, its desirable freshness decreases, while the undesirable aromas characteristic of ageing may increase in concentration and perception [1]. This decline in product quality during its shelf life depends on the storage conditions, which in turn are affected by several factors. Beer ageing is considered one of the most important issues challenging the brewing industry.

Ageing occurs during product storage and involves many chemical reactions that can cause changes in the chemical composition of the beer, thus altering its aroma profile [2]. Primarily, positive aromas, such as fruity and floral aromas, tend to decrease in intensity during the storage period, and aromas such as catty, black-currant, or cardboard along with other aromas such as sweet, caramel, honey, bread, earth, straw, wood, and sherry can arise [3,4]. 

The ageing process is a phenomenon that depends on many factors, such as raw materials and processing conditions, and is greatly affected by the type of packaging used, as well as the storage conditions, such as time, temperature, and light. Appropriate packaging can help delay or reduce these changes in the product during its shelf life, by slowing down the product deterioration, retaining the beneficial effects of processing, extending shelf life, and maintaining the quality and safety of the final product [5,6].

Lorencová et al. [3] evaluated selected physicochemical parameters and organoleptic properties of Czech-type lager beer during a 12-month storage period, concerning the applied type of packaging: glass bottle, aluminium can, polyethylene terephthalate (PET) bottle, and stainless steel beer keg. The study showed that, generally, the type of packaging significantly influenced the physicochemical and organoleptic properties of the examined samples. The authors concluded that the aluminium can and the stainless steel keg were evaluated as the most suitable types of packaging for beer storage and PET the least suitable. The same results were found by Gagula et al. [7] in their study about the influence of packaging material on the volatile compounds of beer.

To monitor the evolution of aroma compounds in beer during its shelf life, natural and forced ageing processes can be used as ageing methods. The forced ageing process is a pervasive but discriminatory method for accelerating the process that occurs during the natural ageing of a beer. Different forcing regimes, involving changes in certain parameters, such as the temperature, time, impact of light, oxygen content, alteration in pH values, and mechanical action (e.g., vibration), can be applied [4]. However, some results from the literature [4,8] indicate that forced ageing alters the aroma profile of beer, unlike natural ageing.

Lehnhardt et al. [8] used sensory and analytical approaches to assess the prediction power of a forced ageing method. To carry out their study, a Pilsen and a Lager beer were stored for up to 17 months at 20 °C (natural ageing) and 40 °C for up to 9 days (forced ageing) and analyzed by gas chromatography olfactometry (GC-O), gas chromatography-mass spectrometry (GC-MS), and sensory analysis with a trained panel. The authors demonstrated that the forced ageing method led to the development of cardboard and bready notes, whereas natural ageing led to fruity and berry notes.

According to Saison et al. [9], the flavor of an aged beer varies significantly depending on the conditions to which it is subjected. In their study, the authors applied different temperature-time profiles, oxidative conditions, and varied pH and ethanol concentrations of the samples. The samples were analyzed by gas-chromatography and sensory analysis. The authors concluded that the ageing process was accelerated when samples were submitted to higher temperatures, to oxidative conditions and, in a lesser degree, to a lower pH. On the other hand, changes in the flavor profile could be observed between samples exposed to different temperatures and oxidative conditions.

Considering consumers’ expectations and knowing that their preferences differ, the type of beer packaging varies significantly between countries. In European countries, for example, the preference is for bottles in the first place, and in second place aluminium cans. Thus, to better understand the issues challenging the brewing industry, such as the ageing process, it is necessary to understand the changes that occur in the volatile matrix of a beer during its shelf life in both types of containers. Therefore, considering the chemical properties of the volatile compounds involved, proper isolation and concentration of the compounds, with an adequate identification with gas chromatography coupled to mass spectrometry (HS-SPME/GC-MS) seems to be the best technique to monitor the evolution of the aroma profile in beer during its natural ageing. Regarding the determination of volatile compounds causing changes during beer natural ageing, Lenhardt et al. [1] compared the results from the different established analytical methods commonly used for that type of analysis, such as headspace solid-phase microextraction (HS-SPME), solvent-assisted flavor evaporation (SAFE), and steam distillation (SD). The article discussed the effect of these methods on flavor stability assessment. The comparison was conducted for four different commercial pale lager beers at different stages of ageing at 20 °C (fresh, 5 months, and 10 months). The results showed that ageing-related changes in pale lager beer presented altered profiles and behavior with SD compared to the non-invasive HS-SPME, due to heat intake. Based on the results presented in the comparison between the analytical methods, the authors indicated that the most gentle and non-invasive method was the best option to apply for analysis. Several reviews and papers are available on the aroma profile of beer; however, there are not many studies on the influence of different packaging types on the aroma profile evolution during natural ageing. In this context, this work aimed to monitor the evolution of the aroma profile of beer packaged in aluminium cans and glass bottles, stored in the absence of light at 14 °C +/− 0.5 °C for a period of 11 months, correlating the sensory data with instrumental data obtained from HS-SPME/GC-MS. We also aimed to determine the main alterations that occurred in the volatile profile of the samples, which allowed classifying the beer according to the container, aluminium can or glass bottle, and the time of storage.

## 2. Results

### 2.1. GC-MS

The GC-MS analysis of beer samples led to the identification of 34 volatile compounds from different chemical families, namely esters, alcohols, acids, ketones, aldehydes, monoterpenes, and phenols. A set of 11 major volatiles were detected in the lager beers: octanoic acid, decanoic acid, caproic acid, 2-phenylethyl alcohol, 2-methyl-1-butanol, 3-methyl-1-butanol, β-phenethyl acetate, isoamyl acetate, ethyl octanoate, ethyl decanoate, and ethyl hexanoate. Table 1 shows the 34 compounds identified using aluminium cans and glass bottles.

### 2.2. Sensory Analysis

The sensory analysis performed was a blind triangle olfactometric test of difference [12], where the untrained assessors were allowed to use only the olfactory sense (nose) to distinguish between samples. The main objective of this test was to differentiate the samples by focusing only on their volatile compounds. A total of 227 individual triangle tests were conducted. To avoid odor saturation of the panelists, only two sessions were held per day. The results were interpreted and analyzed according to the European Brewery Convention Analytica of Sensory Analysis (13.7) [12]. First, triangle tests were performed of beers packaged in aluminium cans and glass bottles, at different periods of their shelf life (fresh from the brewery, fresh from the supermarket, 6 months, and 11 month-aged), as shown in Table 2.

The results from the sensory analysis performed on fresh beer from the brewery (Triangle Test 1) showed that panelists were not able to differentiate between the samples, with only 12 correct responses. The panelists were indeed able to differentiate between beers packaged in aluminium cans and glass bottles in the triangle test performed with samples bought in the supermarket (Triangle Test 2).

Triangle Tests 3 and 4, with 26 and 30 trials, respectively, performed for beers aged 6 and 11 months packaged in aluminium cans and glass bottles, showed that the panelists were able to differentiate between the samples. These results show that, even under optimal conditions of light and temperature, beer can present differences from the sixth month of ageing.

A PCA of the data obtained from the chromatographic analysis revealed that aluminium cans (AC) and glass bottles (GB) could not be distinguished from each other, as shown in Figure 1A.

In Figure 1A, no difference can be seen between the groups of samples (canned or bottled samples). Therefore, we decided to perform triangle tests between the fresh and aged beers in the same container, either aluminium cans or glass bottles. The results are shown in Table 3.

The second group of triangle olfactometric tests (Triangle Tests II) was carried out with samples of fresh beer and aged beer (6 and 11 months), in the same type of containers (aluminium can fresh vs. aluminium can aged, and glass bottle fresh vs. glass bottle aged). For all tests except test 5B, which was performed between fresh and 6-month aged bottled samples, the panelists were able to distinguish fresh from aged beer samples with a 95% level of confidence. These results show that if beer is naturally aged in the absence of light and with a controlled temperature, bottles seem to be the type of container that olfactometrically best preserves the product up to 6 months of ageing.

### 2.3. Chemometric Analysis

The first step in the chemometric analysis was to apply principal component analysis (PCA) to the chromatographic data. The first four PCs explained 73% of the total variance of the model. For a preliminary visualization of the data, we decided to show the score plot of the first two PCs, since they explained 50% of the total variance in the data. Figure 1A,B show the score and loading plots of the PCA analysis for the first two principal components (PCs).

Although the sample size (18 beers) was certainly not large, at least on an exploratory level, some trends were detected. As mentioned above, it was observed that no differences existed between the aluminium cans and glass bottles, although some grouping appeared related to the ageing time. In Figure 1A, the group of fresh beers from the brewery and supermarket show an opposite correlation in PC1 between both types of samples; while the supermarket fresh samples had a negative score on PC1, the brewery samples had a positive one. PC2 explained the difference between the fresh and aged beers. No outlier samples were detected; however, some of the samples had a higher influence on the PCA model. This can be seen in Appendix A, were the Hotelling T2 vs Q residual plots are shown for the PCA models with 1, 2, 3, and 4 PCs. GBFS1, ACFS1, and GBFS2 were the most influential samples. GBFS1 and ACFS1 had a higher leverage (T^2^ value) than the rest of the samples; that is, they appeared at the extreme of all PCA models. However, for the model with four PCs, they were within the limits. Instead, GBFS2 lies in the center of the model (low T^2^ value) but has a higher residual; that is, part of the chromatogram (peak areas) of GBFS2 was different from the rest of samples and was not modelled by PCA. By inspecting the original data matrix, we could observe that GBFS2 had a much lower peak area for octanoic acid.

Figure 1B shows the PCA loading plot. It can be observed that ethyl butyrate (3) and butyl acetate (6) were the variables with most weight on PC2 and showed an opposite sign, indicating that they were negatively correlated. Ethyl butyrate (3), which did not present a significant variation during ageing, had a positive loading in PC2, suggesting that it was positively correlated to fresh samples. Butyl acetate (6), which showed an increase in the peak area during ageing, had a negative loading in PC2, suggesting that it was positively correlated to aged samples.

After the preliminary PCA analysis, we applied partial least squares discriminant analysis (PLS-DA) to the data, to try to discriminate beers depending on the type of container (can/bottle) and the ageing time (fresh/aged). As the number of samples in the training was not set very high, the PLS-DA models built were validated using the leave-one-out cross-validation technique, and the optimal number of latent variables was determined based on the percentage of correctly classified samples for the cross-validation set, as shown in Table 4.

Figure 2 shows the score and loading plots for the first two factors of the PLS-DA model fresh vs. aged samples. In the score plot, it can be observed that the difference between the fresh and aged samples was even more evident than in the score plot of the PCA model (Figure 1A). According to the loading plot, the compounds that helped to characterize samples as fresh were isoamyl acetate (8), ethyl hexanoate (13), hexanoic acid (29), and decanoic acid (34). For the aged samples, the compounds that stood out were 2-methyl-1-butanol (11), 3-methyl-1-butanol (12), linalool (20), β-phenethyl acetate (27), and 2-phenylethyl alcohol (31).

Finally, partial least squares regression (PLSR) was applied to build a model to predict the ageing time of a beer. For this, a regression model using the training set was built between the **X**-matrix (chromatographic peak areas) and a **y**-vector containing the months of ageing of the beers.

Four levels of ageing were used: 0 months (fresh beer from the brewery), 1 month (fresh beer from the supermarket), and 6 and 11 months (aged beers). The model, as for PLS-DA, was leave-one-out cross-validated. In this case, the optimal number of LVs of the model was determined based on the minimum value of the prediction error for the cross-validation set and expressed as the root mean square error of cross validation (RMSECV). Figure 3 shows a plot of the predicted vs. actual values for the validation set and some parameters of the model. The RMSECV value found was around 1.1 months, for a model with seven LVs. This is the average error one could expect when predicting the ageing time of a beer.

## 3. Discussion

During ageing, the beer samples analyzed showed a slight variation in the abundance of some fruity, floral, and sweet aroma compounds. 6-methyl-5-hepten-2-one (15), furfural (18), and β-phenethyl acetate (27) showed an increase in the peak area, while isoamyl acetate (8), linalool (20), ethyl hexanoate (13), and ethyl octanoate (17) showed a decrease in the peak area. For some authors, instead of the formation of new compounds, the changes in aged beer were more related to the variation of the molecules already present in fresh beer, as we observed in the naturally aged samples [13,14,15].

During the natural ageing process, some esters, such as ethyl butyrate (3), ethyl valerate (9), hexyl acetate (14), and ethyl ester octanoic acid (17), showed a decrease in the peak area in both types of container. This decrease occurred due to the ester hydrolysis during ageing [16]. On the contrary, other esters, such as ethyl 3-methylbutanoate (5), isoamyl acetate (8), ethyl hexanoate (13), and ethyl decanoate (23), showed a slight difference between both types of container during the 11 months of storage.

In the study by Vanderhaegen et al. [16], the levels of ethyl 3-methylbutanoate increased in all beers analyzed during the ageing process, due to the reaction of 3-methylbutyric (acid resulting from the degradation of hop bitter compounds) with alcohol. In the same study, isoamyl acetate and ethyl hexanoate showed a decrease from levels below their threshold, diminishing the fruity aroma and consequently decreasing the intensity of the “background” flavor and increasing the perception of eventual stale aromas.

Regarding the sensory analysis, the results of sensory Test 1 between the fresh samples in aluminum cans and glass bottles from the brewery showed that, in the freshest conditions, the panelists were not able to distinguish the samples. Since these samples were transported under the same conditions, kept at a controlled temperature and in the absence of light, and analyzed as soon as possible, to maintain freshness, they should have shown no differences, as observed in the results from test 1 in the sensory analysis.

On the subject of the test performed with fresh samples from the supermarket, Triangle Test 2, the panelists were able to differentiate between aluminum cans and glass bottles. These results could be explained by the different storage conditions imposed on these samples compared to the fresh samples from the brewery. The fresh samples from the brewery were delivered directly from the brewery after packaging and were kept in darkness and at a controlled temperature of 14 °C +/− 0.5 °C. The samples purchased from the supermarket had been packaged approximately one month previously and stored under supermarket conditions of light and temperature. Many studies have proven that an increase in the storage temperature generally has a negative impact on beer quality and stability, due to the degradation of iso-α-acids and the deterioration of aroma (an increase of staling compounds, beer color, and haze formation) [17,18,19]. According to Paternoster et al. [20], since high temperatures can increase the energy that enables ageing reactions, exposing beer to different temperatures can lead to the formation or degradation of metabolites, which can activate or deactivate various ageing reactions, leading to different sensory attributes. Additionally, the exposure to light could contribute to accelerating the deterioration rate of the glass bottled samples.

Moreover, by inspecting the chromatogram data, we could observe that the peak area of furfural (18) in the fresh beers purchased in the supermarket was higher than that of the fresh samples from the brewery. For some authors, furfural can be considered a heat indicator during the ageing process [9,21]. In addition, according to Saison et al. [22], despite its low threshold, furfural has been used as an ageing indicator, due to its close correlation with sensory scores of flavor staling. If the supermarket samples were exposed to variations in temperature during transportation or storage, this could have caused a change in the volatile profile of the samples that would allow the panelists to differentiate these samples from the fresh samples from the brewery.

The sensory tests applied to samples aged for 6 and 11 months, Triangle Tests 3 and 4, showed that, for the panelists, these samples were olfactometrically different. In the study performed by Lorencovà et al. [3] with forced aging, the results of the sensory analysis between different types of packaging showed that the samples stored in aluminum cans were the best evaluated by the panel of expert assessors. According to the authors, this type of packaging is capable of retaining all organoleptic characteristics, and showing a weaker fading and a slight increase in bitterness after 10 months of storage. This could be explained by the oxygen contained in the beer, both the oxygen dissolved in the liquid and the oxygen present in the headspace. According to studies related to the oxygen content of beer, it decreases during transportation, which could be the result of oxidative reactions, which contribute to the deterioration of the aroma quality during storage [13,23,24].

Unlike what is discussed in the previous paragraph, the results obtained from the Triangle tests II, carried out with samples in the same type of container, showed that, for naturally aged beers under optimal conditions of light and temperature, bottles seem to be a better container than a cans, since the panelists were not able to detect a difference between fresh and 6 month aged bottled samples, which was not the case with samples packed in aluminum cans.

According to Onaliran et al. (2017) [25] and Saison et al. (2018) [9], during storage, the sensory quality of the product tends to deteriorate significantly over time, at a rate that depends on the beer composition and storage conditions. As mentioned previously, the time of storage, temperature, oxygen, and light are important contributors to the degradation of aroma quality. Even under optimal storage conditions, beer quality deteriorates significantly as the product approaches the expiration date, regardless of the type of container in which it is packed.

Furthermore, when analyzing and comparing the GC-MS data of the beers contained in aluminum cans and glass bottles, and their evolution during ageing using chemometric tools, the variation shown in the peak areas of certain compounds listed above was not sufficient to instrumentally determine that the samples were different based on the type of packaging. Figure 1A and Figure 2 show that the samples were divided in two main groups: fresh and aged. This could be verified by the results of the PLS-DA in Table 4, which better classified the samples as fresh and aged.

The aroma compounds responsible for the classification of the samples into fresh and aged were: isoamyl acetate (banana), ethyl hexanoate (apple peel, fruit), caproic acid (sweet) and decanoic acid (rancid, fat); and, 2-methyl-1-butanol (malt), 3-methyl-1-butanol (malt), linalool (flower, lavender), β-phenethyl acetate (rose, tobacco, honey), and 2-phenylethyl alcohol (honey, spice, lilac). During the ageing process beer tends to show a decrease in bitterness, which is partly explained by the sensory masking effect produced by an increasing sweet aroma [26]. Additionally, positive fruity/estery aromas that come from compounds such as isoamyl acetate tend to decrease in intensity.

Finally, the PLS-R model built was capable of predicting the ageing period of lager beer (fresh, 1 month, 6 months, and 11 months aged), with an error of 1.1 month.

## 4. Materials and Methods

### 4.1. Beer Samples

One hundred samples of a commercial lager beer with an alcohol content of 5.4% *v*/*v*, 50 packaged in glass bottles and 50 packaged in aluminum cans, were delivered directly from a local brewery, in the freshest possible conditions, and used for the controlled natural ageing experiment. The samples were aged in the absence of light at 14 °C +/− 0.5 °C for 11 months. A total of 41 samples were used for the sensory analysis and 18 samples were used for the GC-MS analysis.

Thirty samples of the same beer brand were purchased in the freshest possible conditions (less than 1 month of packaging) in a local supermarket in both packages: glass bottles and aluminum cans. The samples were used in the sensory analysis, to verify if the storage conditions imposed in the supermarket affected the sensory attributes of fresh beer. All beer samples were degassed by ultrasonication for 15 min prior to GC-MS analysis. Samples were analyzed in triplicate.

### 4.2. Headspace Solid-Phase Microextraction (HS-SPME)

The SPME holder, for manual sampling, and the divinylbenzene-carboxen-polydimethylsiloxane (DVB/CAR/PDMS) 50/30 μm fiber used in this investigation were purchased from Supelco (Bellefonte, PA, USA). All fibers were conditioned prior to use and thermally cleaned between analyses by inserting them into the GC injection port at the temperature recommended by the manufacturer.

All beer samples were degassed by ultrasonication for 15 min before the analysis. The optimal conditions that allowed the extraction of the largest number of odorants and with the highest intensity were achieved by placing 10 mL of sample into a 20 mL glass vial with 3.2 g of NaCl (saturation) and a small magnetic stir bar (the extraction was carried out under constant magnetic stirring). The vials were hermetically capped with a silicon septum under N_2_ atmosphere and were pre-equilibrated for 10 min at 40 °C in a thermostatic bath. Then, a solid phase micro extraction (SPME) device was manually pushed through the vial septum and the fiber was exposed to the headspace vial for 1 h at 40 °C. Afterwards, the fiber was pulled into the needle assembly and the SPME device was removed from the vial. Finally, the fiber was inserted into the injection port for thermal desorption of the analytes at 270 °C for 1 min in the splitless mode of the gas chromatograph.

### 4.3. Gas Chromatographic Analysis

Samples were analyzed with a GC-MS equipment from Agilent Technologies (Santa Clara, CA, USA). This was composed of a 7890 gas chromatograph and a 5977B HES mass spectrometric detector equipped with a high-efficiency ion source. To carry out the chromatographic separations a Chrompark (Varian, Middelburg, The Netherlands) CP-WAX 57 CB (50 m × 0,25 mm i.d., 0.2 µm film thickness) fused silica capillary column was employed. The oven temperature was programmed as follows: the initial temperature was 40 °C; after 5 min it was raised a rate of 3.5 °C/min to 120 °C, and finally a rate of 5 °C/min to 215 °C and held for 10 min. The split–splitless injection port was operated in splitless mode at 270 °C for 1 min. The mass spectra were recorded by electronic impact (EI) ionization at 70 eV with a temperature of 230 °C in the ion source and 150 °C in the mass quadrupole. The mass range analyzed was from 35 to 300 amu (atomic mass units).

### 4.4. Compound Identification

The odorants detected were identified using an Automatic Mass Spectral Deconvolution and Identification System (AMDIS) using a library of mass spectral databases (NIST MS Search version 2.3) by comparison with reference substances based on the retention index (RI) in the website of Flavornet [10], on two stationary phases of different polarity (CP-WAX 57CB and HP-5 MS). To calculate these RI values, a series of *n*-alkanes (from 8 to 20 carbon atoms) were injected under the same chromatographic conditions.

### 4.5. Sensory Analysis

The beer samples were sensory evaluated by a panel of untrained assessors, aged between 18 and 50 years, and the tests were performed following the methodology and statistics of the European Brewery Convention [12]. The assessors performed two triangle olfactometric tests of difference to determine whether the samples were significantly different: (i) between aluminum cans and glass bottles, at different periods of shelf life (fresh, 6-months, and 11-months), and (ii) between the same beer samples without differentiating the type of container, comparing fresh samples and samples aged for 6 and 11 months. For the evaluation, samples were served (50 mL) in black glasses (300 mL; coded with 3-digit numbers) that were odorless and covered with watch glasses. The samples were served in a sensory laboratory in random order and under normal light and temperature conditions. The assessors were presented with a set of three coded samples, two of which were identical. Assessors were asked to identify the olfactometrically different sample. As this is a forced-choice method, if assessors could not identify a difference, they had to make a guess. The results were interpreted and analyzed according to the European Brewery Convention Analytica of Sensory Analysis (13.7) and the significance level used was ≤0.05 [12].

### 4.6. Chemometric Methods

The chromatographic profiles collected on the samples were processed using chemometric classification and prediction methods. The data obtained were exported to an Excel table and structured in a matrix of dimensions 18 samples × 34 columns (34 identified compounds). The chromatographic values of the matrix were the integrated peak areas of each identified compound. Before the chemometric analysis, the matrix values were pre- processed, since the difference between the high and low peaks was significant, affecting the analysis. The pre-processing of the values was based on the selection of a representative peak of the high peaks and one of the low peaks, with the smallest relative deviation in all samples. The high peak chosen was 2-phenylethyl alcohol (31), and the low peak selected was p-vinylguaiacol (33), with a median relative deviation of 12% for both compounds. Then, the values of the peak area of the high peaks in the matrix were divided by p-vinylguaiacol (33) and of the low peaks by 2-phenylethyl alcohol (31), creating a new matrix based on the original matrix. As no significant deviations were found in the data, only natural differences in the peak areas, the only pre-processing applied to the data was autoscaling (i.e., mean centering and standardization to unit variance).

PCA was used for a preliminary visualization of the GC-MS data. PCA reduces the dimensionality of a data set by finding an alternative set of variables, called principal components (PCs), which retain most of the information contained in the original data. Each PC is a linear combination of the original variables and is orthogonal to each other. The relationship between samples, variables, and sample/variables was revealed when scores and loadings were plotted [27]. PCA can reveal groups and trends in the data and point out outlier samples.

PLS-DA was used to discriminate beer samples depending on the type of container (aluminum can or glass bottle) and to discriminate fresh beers from aged beers. PLS-DA is a discriminant method that is based on the partial least squares regression (PLSR) algorithm, as described below. To classify the samples, a PLS-DA model was built by correlating the matrix **X** of predictor variables (peak areasin this case) with a vector **y** of dummy variables, zeros and ones in a two-class problem, as in this work. In the first case, the value 0 was assigned to the aluminum cans and the value 1 to the glass bottles. In the second case, the value 0 was assigned to fresh beers and the value 1 to aged beers. Then, a PLS model was built between the experimental matrix **X** and the binary-coded vector **y**. For new samples, the predicted values were distributed around zero and one, and a threshold is a set to assign the samples to a given class.

Finally, partial least squares regression (PLSR) was used to model and predict the ageing time of the samples. PLSR is a multivariate calibration method that correlates a matrix **X** of predictor variables (peak areas in our case) with a vector **y** containing the property of interest (in this study the ageing time) [14]. Four levels of ageing were considered: 0 months (fresh from the brewery), 1 month (fresh from the supermarket), 6 months, and 11 months.

All calculations were performed using PLS Toolbox v8.7 (Eigenvector Research Inc., Eaglerock, LA, USA) running with MATLAB R2021a (The MathWorks, Natick, MA, USA).

## 5. Conclusions

In this paper, the effect of container and time of ageing of a beer stored under optimal conditions was sensorially and instrumentally monitored and evaluated. In general, the type of packaging influenced the olfactometric perception by the panelists, who were able to differentiate between canned and bottled samples at all ageing times analyzed, except the fresh samples from the brewery. However, instrumentally, the samples could not be differentiated by the type of packaging, but only by the ageing time. The olfactometric difference could be explained by the presence of varied esters in the sample, which could have interfered with the aroma profile of the beer, due to the synergistic effect that these esters have on individual flavors, which means that a slight variation in their concentration may have had a critical effect on the organoleptic perception of the product. On the other hand, the instrumental variation of the peak areas of the compounds in both types of container was not as marked as the variation presented for the ageing time.

Despite the number of samples not being high (18 samples), it can be said that at an exploratory level, it was possible to detect some trends in the samples. In a general way, we can say that multivariate analysis proved to be a useful tool for discriminating beer samples based on the time of storage (fresh from the brewery, 1 month from the supermarket, 6 month-aged, and 11-month-aged) but not for discriminating by packaging type (aluminum cans or glass bottles). PLS-DA showed that the samples could be classified into two groups: fresh and aged. Finally, PLSR was able to relate the chromatographic peak areas with the ageing time (fresh from the brewery, 1 month from the supermarket, 6-month aged, and 11-month-aged) and predict the ageing time with an error of 1.1 months. To strengthen the models presented by the chemometric analysis and obtain conclusive results, it would be necessary to analyze more samples.

Further studies, including a descriptive sensory analysis by a trained panel, could lead to a better understanding of the most important differences that occurred during the natural ageing process and that led to a sensory difference between the canned and bottled beers. Additionally, a better understanding of the ageing process could be achieved by applying other chromatographic techniques, such as gas chromatography–olfactometry (GC–O) or improving the HS-SPME/GC-MS technique. This would make it possible to quantify the aromatic compounds related to the ageing process, thus being able to determine with greater precision the ageing marker compounds for each type of container.

In conclusion, the combination of sensory, GC-MS, and multivariate analyses seems to be a valuable tool for discriminating beer samples at different periods of shelf-life and could be used for monitoring and identifying possible changes in the volatile fraction during ageing.

## Figures and Tables

**Figure 1 molecules-28-02807-f001:**
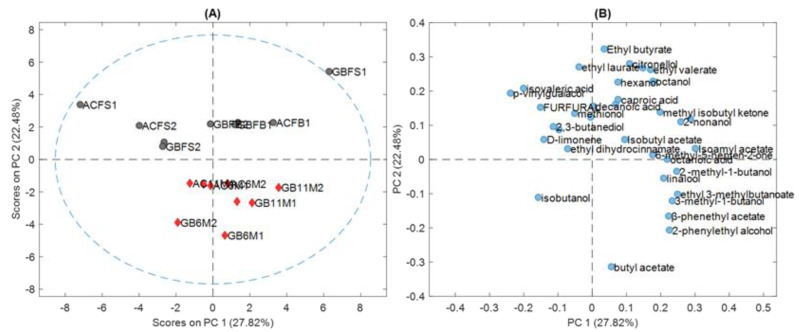
(**A**) Score plot of PCA analysis; (**B**) Loading plot of PCA analysis. ACFB—aluminium can fresh brewery; GBFB—glass bottle fresh brewery; ACFS—aluminium can fresh supermarket; GBFS—glass bottle fresh supermarket; AC6M—aluminium can 6 months aged brewery; GB6M—glass bottle 6 months aged brewery; AC11M—aluminium can 11 months aged brewery; GB11M—glass bottle 11 months aged brewery. Grey circles indicate fresh beers, red triangles aged beers and blue circles names of volatile compounds identified.

**Figure 2 molecules-28-02807-f002:**
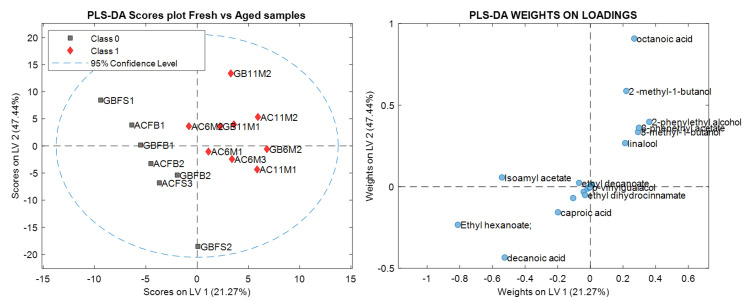
Score and loading plots for the first two factors of the PLS-DA model fresh vs. aged. Grey circles indicate fresh beers, red triangles aged beers and blue circles names of volatile compounds identified.

**Figure 3 molecules-28-02807-f003:**
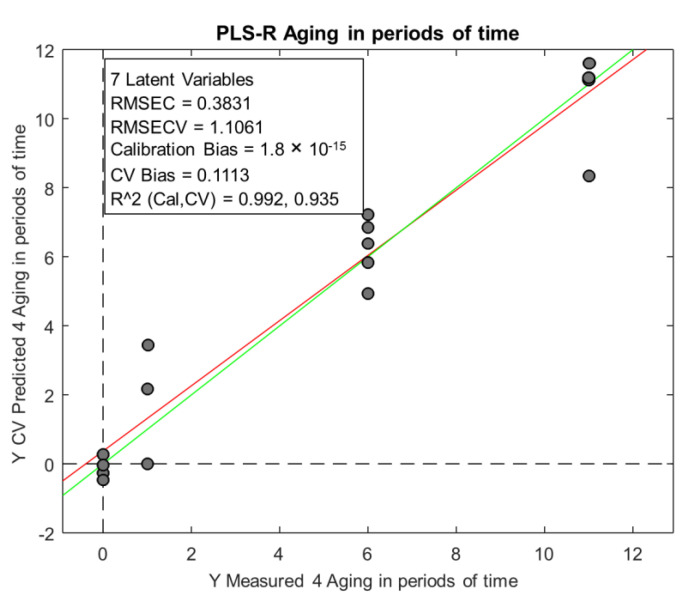
Plot of predicted vs. actual values for the validation set.

**Table 1 molecules-28-02807-t001:** GC-MS Compound identification during beer ageing under optimal storage conditions.

Number	CAS Number	Retention Index * [10]	Compound Name	Odor Impression [10]	Threshold [11]	Class Compound
**01**	108-10-1	969	Methyl isobutyl ketone	—	240 to 640 ppb	Ketone
**02**	110-19-0	1014	Isobutyl acetate	Fruit, apple, banana	65 to 880 ppb	Ester
**03**	105-54-4	1024	Ethyl butyrate	Apple	0.1 to 18 ppb	Ester
**04**	71-23-8	1039	Propanol	Alcohol, pungent	5.7 to 40 ppm	Alcohol
**05**	108-64-5	1055	Ethyl 3-methylbutanoate	Fruit	NA	Ester
**06**	123-86-4	1058	Butyl acetate	Pear	10 to 500 ppb	Ester
**07**	78-83-1	1104	Isobutanol	Wine, solvent, bitter	360 ppb to 3.3 ppm	Alcohol
**08**	123-92-2	1123	Isoamyl acetate	Banana	2 to 43 ppb	Ester
**09**	539-82-2	1130	Ethyl valerate	Yeast, fruit	1.5 to 5 ppb	Ester
**10**	5989-27-5	1172	D-limonene	Citrus, mint	NA	Monoterpene
**11**	137-32-6	1209	2 -methyl-1-butanol	Malt	0.14 mg/L	Alcohol
**12**	123-51-3	1213	3-methyl-1-butanol	Whiskey, malt, burnt	250 ppb to 4.1 ppm	Alcohol
**13**	123-66-0	1222	Ethyl hexanoate	Apple peel, fruit	0.3 to 5 ppb	Ester
**14**	142-92-7	1247	Hexyl acetate	Fruit, herb	2 to 480 ppb	Ester
**15**	110-93-0	1320	6-methyl-5-hepten-2-one	Herb, butter, resin	50 ppb	ketone
**16**	111-27-3	1342	Hexanol	Resin, flower, green	200 ppb to 2.5 ppm	Alcohol
**17**	106-32-1	1422	Ethyl octanoate	Fruit, fat	5 to 92 ppb	Ester
**18**	98-01-1	1446	Furfural	Caramel, bready, cooked meat	280 ppb to 8 ppm	Aldehyde
**19**	628-99-9	1521	2-nonanol	Cucumber	52 to 82 ppb	Higher alcohol
**20**	78-70-6	1544	Linalool	Flower, lavender	4 to 10 ppb	Monoterpene alcohol
**21**	111-87-5	1557	Octanol	Chemical, metal, burnt	42 to 480 ppb	Alcohol
**22**	513-85-9	1602	2,3-butanediol	Fruit, onion	NA	Alcohol
**23**	110-38-3	1629	Ethyl decanoate	Grape	8 to 12 ppb	Ester
**24**	503-74-2	1660	Isovaleric acid	Sweet, acid, rancid	190 ppb to 2.8 ppm	Acid
**25**	505-10-2	1730	Methionol	Sweet, potato	0.2 ppb	Alkyl sulfide
**26**	106-22-9	1748	Citronellol	Rose	11 ppb to 2.2 ppm	Monoterpene
**27**	103-45-7	1815	β-phenethyl acetate;	Rose, honey, tobacco	3.8 ppm	Ester
**28**	106-33-2	1827	Ethyl laurate	Leaf	NA	Ester
**29**	142-62-1	1830	Hexanoic acid	Sweet	93 ppb to 10 ppm	Fatty Acid
**30**	2021-28-5	1902	Ethyl dihydrocinnamate	Flower	17 to 40 ppb	Ester
**31**	60-12-8	1918	2-phenylethyl alcohol	Honey, spice, rose, lilac	0.015 ppb to 3.5 ppm	Alcohol
**32**	124-07-2	2035	Octanoic acid	Sweet, cheese	910 ppb to 19 ppm	Fatty Acid
**33**	7786-61-0	2203	p-vinylguaiacol	Clove, curry	0.75 to 3 ppb	Phenol
**34**	334-48-5	2229	Decanoic acid	Rancid, fat	2.2 to 10 ppm	Fatty acid

* Retention index for a polar column. NA: not available.

**Table 2 molecules-28-02807-t002:** Results from the Triangle Olfactometric Test I with α ≤ 0.05.

Triangle Test I: Aluminium Can and Glass Bottle					
Triangle Test	Trials ᵃ	Successes ᵇ	*p*-Value	Minimum Number of Correct Responses	Results
**1**	28	12	0.1911	15	ND
**2**	30	16	0.0188	15	D
**3**	26	14	0.0247	14	D
**4**	30	18	0.0025	15	D

**Triangle Test I: 1** = Fresh aluminium can and fresh glass bottle from brewery; **2** = Fresh aluminium can and fresh glass bottle from supermarket; **3** = 6-month aged aluminium can and 6-month aged glass bottle from brewery; **4** = 11-month aged aluminium can and 11-month aged glass bottle from brewery. ᵃ Number of participants; ᵇ Number of correct responses; ND = No difference between samples; D = Samples are different.

**Table 3 molecules-28-02807-t003:** Results from the Triangle Olfactometric Test II with α ≤ 0.05.

Triangle Test II: Fresh and Aged Beer without Differentiate Types of Packaging (6 Months and 11 Months)					
Triangle Test	Trials ᵃ	Successes ᵇ	*p*-Value	Minimum Number of Correct Responses	Results
**5A**	26	14	0.0247	14	D
**5B**	28	12	0.1911	15	ND
**6A**	30	16	0.0188	15	D
**6B**	29	17	0.0045	15	D

**Triangle Test II**: **5A** = Aluminium can fresh and aluminium can aged for 6 months; **5B** = Glass bottle fresh and glass bottle aged for 6 months; **6A** = Aluminium can fresh and aluminium can aged for 11 months; **6B** = Glass bottle fresh and glass bottle aged for 11 months. ᵃ Number of participants; ᵇ Number of correct responses; ND = No difference between samples; D = Samples are different.

**Table 4 molecules-28-02807-t004:** Results of the PLS-DA models can vs. bottle and fresh vs. aged beer.

Model	n° LVs	% Accuracy	% Sensitivity	% Specificity
Can vs. bottle	4	56	67	50
Fresh vs. aged	4	100	100	100

## Data Availability

The data presented in this study are available on request from the corresponding author.

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
