# Peer review of "Monitoring the Evolution of the Aroma Profile of Lager Beer in Aluminium Cans and Glass Bottles during the Natural Ageing Process by Means of HS-SPME/GC-MS and Multivariate Analysis"

_molecules, 2023, doi:10.3390/molecules28062807_

Round 1

Reviewer 1 Report

The chemical composition of beers changes in course of their shelf life and this fact sometimes predetermines consumer experience. Though a lot of works describing aging of beer have already been published the complexity of the problem remains to be a challenge for the researchers. For these points I consider the manuscript worthy of publishing. Nevertheless some points of it should be clarified:

1.       The comparison of components content is performed using absolute values of areas of the corresponding peaks without any quantitations or even internal standard calculations. Why? Are these areas reproducible? How many repetitions were done for every analysis?

2.       There is not any description of MS detection method except used mass range.

3.       The authors mentioned NIST Mass Spectral Library version 2.2 but I suppose the mean MSSearch version. Please check and correct.

Reviewer 2 Report

This study is interesting and deserves to be published, but some paragraphs need to be detailed and improved.

The paragraph on the interpretation of the graphs of the principal component analysis is not clear and includes some misinterpretations.

Care must be taken when performing a PCA with tables that have fewer rows than columns (18 samples x 34 variables). A sample of size n with p dimensions has at most n−1 principal components if n <= p. So, it is possible to derive principal components when you have fewer samples than dimensions, but only n−1 components. In any case, this is probably not desirable, because the sample is too small to derive a fully meaningful transform.

The authors do mention that principal components 1 and 2 explain only about 50% of the total variance in the data, but it should also be said that the interpretation made in the PC1-PC2 design can only be partial. In addition, some individuals and variables are poorly represented in this design. Not all points near the centre can be interpreted.

In this respect, it would be interesting to have the values of the squared cosines and contributions (in appendix) of all the points, especially the individuals. The individuals that seem to be best represented are ACFS1, GBFS and to a lesser extent GB6M2 and GB6M1. For the others, not much can be said, as they are poorly represented.

Why are GBFS1 and GBFS2 so far apart when they appear to be replicas? It is then difficult to draw conclusions about these beer types. Is GBFS2 an outlier?

The most obvious difference is the difference between fresh beers and beers aged 6 months on axis 2

It is written: “The group of aged beers (11 and 6 months) is more compact than the group of fresh beers, which shows more dispersion along PC1. PC2 explain the difference between fresh and aged beers”. The sentence is not very clear, the majority of the points are misrepresented, and the scatter is especially large along axis 1 for the fresh beer group

Figure 1 is redundant with Figure 2 (A). The use of colour and different point types does not help the reader, on the contrary.

The interpretation of figure 3 is not clear either, the most influential molecules are mainly ethyl ester octanoic acid, 3-methyl-1-butanol and 2-phenylethyl a (?). The figure is cut off.

The conclusions on the PLS-DA and PLS-R analyses are also to be moderated due to the small number of samples compared to the number of variables. This article is very exploratory, and the analyses should be strengthened.

Round 2

Reviewer 1 Report

The author have made all necessary corrections and the manuscript can be published in my opinion. The only point which can be improved is the substitution of term 'electron impact' by 'electron ionization' as it recommended by IUPAC (https://goldbook.iupac.org/terms/view/E01999)

Reviewer 2 Report

Authors have made great efforts to enhance the manuscript.

All points have been well treated and questions from the reviewers were answered.

This version of the manuscript is improved now.